# Wavelet Subband Alignment for Continual Test-Time Adaptation in Video Action Recognition

## Abstract

Robust action recognition in the wild is challenged by non-stationary corruptions in streaming video that shift the test distribution over time. We propose Wavelet Subband Alignment (WSA), a parameter-free method for continual test-time adaptation (CTTA) that aligns frequency-aware feature statistics at inference. WSA applies a 2D/3D discrete wavelet transform (DWT) to normalization-layer features and aligns per-channel first/second moments of the base branch and subbands to precomputed source (clean) statistics via online exponential moving averages. The method adds no layers, performs no reconstruction, and naturally extends to 3D time–space subbands to address temporal drift. On UCF101 and Something-Something v2 across twelve corruption families, WSA consistently outperforms strong TTA/CTTA baselines under continual and random shift protocols, remains robust when in-domain statistics are unavailable (cross-dataset transfer), and redistributes energy from coarse low frequency toward informative higher-frequency subbands. These results establish frequency-aware subband moment alignment as an effective mechanism for stabilizing inference on evolving video streams.

## 1 Introduction

Action recognition systems in the wild—e.g., surveillance, sports, and mobile/egocentric video—face non-stationary corruptions (noise, blur, compression, weather, contrast) that drift over time and degrade inference. While image TTA is well studied [6, 16, 19, 22], video-centric TTA is required to exploit temporal structure and remain stable on evolving streams [10, 28]. Continual TTA (CTTA) further seeks online adaptation without source data [17, 23], yet many methods remain image-oriented and overlook the spatiotemporal frequency structure of video corruptions. We therefore target CTTA for action recognition on Something-Something v2 (SSv2) [3] and UCF101 [20], aiming for robust streaming performance with minimal memory.

We propose Wavelet Subband Alignment (WSA), a parameter-free CTTA plugin that aligns base (spatial) and wavelet (frequency) subband statistics of intermediate normalization features to precomputed source (clean) statistics. WSA applies a 2D/3D discrete wavelet transform to normalization features, aggregates per-band moments, and updates them online via exponential moving averages; treating the previous EMA state as fixed avoids backpropagation-through-time. WSA3D aligns time–height–width subbands, and WSA2D aligns spatial subbands. Unlike wavelet architectures that learn invertible downsampling or reconstruct with IDWT [7, 27, 29], WSA leaves the backbone unchanged, uses DWT only as an analysis operator, introduces no new layers, and complements simple temporal resampling and consistency across views [10].

Empirically, WSA consistently redistributes 3D–DWT energy in intermediate features away from the coarse low-frequency LLL component toward higher-frequency spatiotemporal subbands across corruptions (Figure 1), yielding sharper, more detail-attentive representations consistent with

frequency-domain perspectives [1, 7, 9, 18, 25–27, 29]. Evaluated on UCF101 and SSv2 with twelve corruption families, and on TANet [14] and Video Swin [15], WSA improves over non-adaptive and prior TTA/CTTA baselines under continual and random shifts. It requires no source samples at test time (only once-computed clean statistics) and can reuse statistics from another dataset when in-domain data are unavailable (cf. Table 3).

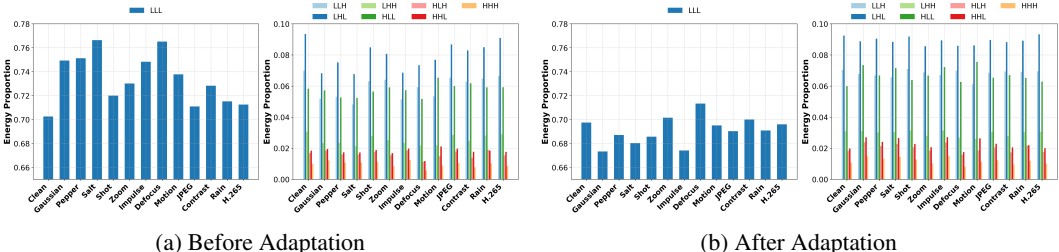

(a) Before Adaptation                    (b) After Adaptation

Figure 1: **Effect of WSA.** 3D–DWT subband energy (TANet, UCF101): 1a before and 1b after adaptation. WSA consistently redistributes energy from the coarse low-frequency LLL subband toward higher-frequency spatiotemporal subbands across corruptions, yielding sharper, more detail-attentive features. L/H denote low/high along (T,H,W).

## 2 Methods

Let $\mathcal{X} \in \mathbb{R}^{N \times C \times T \times H \times W}$ denote a batch of feature tensors extracted from a normalization layer, where $N$ is the batch size, $C$ is the number of channels, and $(T, H, W)$ denote the temporal, height, and width dimensions, respectively.

### 2.1 Wavelet Subband Alignment (WSA)

Features tapped from normalization layers are transformed by a 2D/3D DWT into a base branch and four subband groups, $s \in \{s_1, s_2, s_3, s_4\}$. The alignment module computes per-channel first/second moments for the base and each subband and aligns them to precomputed source (clean) statistics using online exponential moving averages (EMAs).

We employ the Haar wavelet transform for its efficiency and ability to capture multi-scale structure. The Haar analysis filters are $h_L = \frac{1}{\sqrt{2}}[1, \ 1]$ (low-pass) and $h_H = \frac{1}{\sqrt{2}}[1, \ -1]$ (high-pass).

For an input tensor $x \in \mathbb{R}^{n_1 \times \cdots \times n_d}$, a $d$-dimensional DWT ($d \in \{2, 3\}$) produces $2^d$ subbands. We group them into four frequency bands $s \in \{s_1, s_2, s_3, s_4\}$. For $d = 2$, the bands are $s_1 = \{\text{LL}\}$, $s_2 = \{\text{LH}\}$, $s_3 = \{\text{HL}\}$, and $s_4 = \{\text{HH}\}$. For $d = 3$, the bands are $s_1 = \{\text{LLL}, \text{HLL}\}$, $s_2 = \{\text{LLH}, \text{HLH}\}$, $s_3 = \{\text{LHL}, \text{HHL}\}$, and $s_4 = \{\text{LHH}, \text{HHH}\}$.

Let $\alpha = (a_1, \ldots, a_d)$ denote a subband, with $a_k \in \{L, H\}$ selecting the low/high filter along dimension $k$. The subband coefficients are obtained by using the DWT equation,

$$y_\alpha[o_1, \ldots, o_d] = \sum_{i_1, \ldots, i_d} \Big( \prod_{k=1}^{d} h_{a_k}[i_k] \Big) \times x[2o_1 - i_1, \ldots, 2o_d - i_d]. \tag{1}$$

We apply the DWT independently per sample and per channel; we write $y_\alpha[b, c, \cdot]$ to denote the coefficients of subband $\alpha$ for sample $b$ and channel $c$ and then define the band as the collection of its subbands: $y_s = \{y_\alpha \mid \alpha \in s\}$. For convenience, let $K_s = |s|$ denote the number of subbands in band $s$ ($K_s = 1$ for $d = 2$; $K_s = 2$ for all $s$ when $d = 3$ under the above grouping). Let $\Omega_d$ denote the per-subband coefficient index set after stride-2 downsampling in each transformed dimension: its size is $|\Omega_d| = \frac{H}{2} \cdot \frac{W}{2}$ for $d = 2$, and $|\Omega_d| = \frac{T}{2} \cdot \frac{H}{2} \cdot \frac{W}{2}$ for $d = 3$. We pool band-wise by aggregating all coefficients across the subbands in $s$. The total number of coefficients per sample, per channel, in band $s$ is $|\mathcal{U}_s| = K_s \cdot |\Omega_d|$.

 **2.1.1 Base and Subband Statistics**

68 We use a flattened index throughout for consistency. Define the base index set $\Omega_{\text{base}}$ with $|\Omega_{\text{base}}| =$
69 $M = T \cdot H \cdot W$. (Equivalently, $\Omega_{\text{base}} = \{(t, i, j)\}$ with $t \in [1..T]$, $i \in [1..H]$, $j \in [1..W]$, flattened
70 to a single index $u$.) For each channel $c$:

$$\mu_{\text{base}}^c = \frac{1}{N\,M} \sum_{b=1}^{N} \sum_{u \in \Omega_{\text{base}}} \mathcal{X}[b, c, u], \tag{2}$$

$$\nu_{\text{base}}^c = \frac{1}{N\,M} \sum_{b=1}^{N} \sum_{u \in \Omega_{\text{base}}} \big(\mathcal{X}[b, c, u] - \mu_{\text{base}}^c\big)^2. \tag{3}$$

71 Similarly, on subband statistics: For each band $s$ and channel $c$, let $|\mathcal{U}_s| = K_s\,|\Omega_d|$, and define a
72 residual index set $\Omega_{\mathcal{R}}$ of size $|\Omega_{\mathcal{R}}| = \begin{cases} T, & d = 2, \\ 1, & d = 3. \end{cases}$ Then

$$\mu_s^c = \frac{1}{N\,|\mathcal{U}_s|\,|\Omega_{\mathcal{R}}|} \sum_{b=1}^{N} \sum_{\alpha \in s} \sum_{r \in \Omega_{\mathcal{R}}} \sum_{u \in \Omega_d} y_\alpha[b, c, r, u], \tag{4}$$

$$\nu_s^c = \frac{1}{N\,|\mathcal{U}_s|\,|\Omega_{\mathcal{R}}|} \sum_{b=1}^{N} \sum_{\alpha \in s} \sum_{r \in \Omega_{\mathcal{R}}} \sum_{u \in \Omega_d} \big(y_\alpha[b, c, r, u] - \mu_s^c\big)^2 \tag{5}$$

73 **Source-domain statistics.** We denote by $\{\hat{\mu}_{\text{base}}^c, \hat{\nu}_{\text{base}}^c\}$ and $\{\hat{\mu}_s^c, \hat{\nu}_s^c\}_{s \in \mathcal{S}}$ the source-domain (train-
74 ing) statistics for the base and each band, estimated on the source data (e.g., precomputed dataset
75 means/variances).

## 2.2 Continual Adaptation

77 The alignment objective is defined as the $L_1$ distance between target-domain and source-domain
78 statistics. Direct estimation from a single element $\mathcal{X}_i$ is unstable, so we employ exponential moving
79 averages (EMAs). For iteration $i$ and momentum $1-\alpha=0.9$, we update $\mu_{\text{base}}^{c,(i)} = \alpha\,\mu_{\text{base}}^c(\mathcal{X}_i) +$
80 $(1-\alpha)\,\mu_{\text{base}}^{c,(i-1)}$, $\nu_{\text{base}}^{c,(i)} = \alpha\,\nu_{\text{base}}^c(\mathcal{X}_i) + (1-\alpha)\,\nu_{\text{base}}^{c,(i-1)}$, $\mu_s^{c,(i)} = \alpha\,\mu_s^c(\mathcal{X}_i) + (1-\alpha)\,\mu_s^{c,(i-1)}$, and
81 $\nu_s^{c,(i)} = \alpha\,\nu_s^c(\mathcal{X}_i) + (1-\alpha)\,\nu_s^{c,(i-1)}$.

82 The per-iteration loss used to guide adaptation is

$$\begin{aligned}
\mathcal{L}_{\text{WSA}}^{(i)} = \lambda_b \sum_{c=1}^{C} \Big( |\mu_{\text{base}}^{c,(i)} - \hat{\mu}_{\text{base}}^c| + |\nu_{\text{base}}^{c,(i)} - \hat{\nu}_{\text{base}}^c| \Big) \\
+ \sum_{s \in \mathcal{S}} \lambda_s \sum_{c=1}^{C} \Big( |\mu_s^{c,(i)} - \hat{\mu}_s^c| + |\nu_s^{c,(i)} - \hat{\nu}_s^c| \Big).
\end{aligned} \tag{6}$$

83 where $s \in \{s_1, s_2, s_3, s_4\}$ indexes the four frequency bands.

## 2.3 Aggregated Temporal Perturbations

85 We adopt the temporal augmentation scheme of [10], producing $M$ temporally resampled versions of
86 each input $\mathcal{X}_i$, denoted $\{\mathcal{X}_i^{(m)}\}_{m=1}^{M}$. To compute the statistics used in the online EMA—$\mu_{\text{base}}^c(\mathcal{X}_i)$,
87 $\nu_{\text{base}}^c(\mathcal{X}_i)$ and, for every $s \in \mathcal{S}$, $\mu_s^c(\mathcal{X}_i)$, $\nu_s^c(\mathcal{X}_i)$—we replace the single-view calculation with a
88 uniform average over all $M$ temporal resamplings (with the same indices as specified earlier). These
89 averaged estimates are then propagated through the EMA updates without modification.

90 In addition, we encourage prediction invariance across the temporal views. With $f(\cdot\,; \theta)$ denoting
91 the network prediction function, we construct the pseudo-target $y(\mathcal{X}_i) = \frac{1}{M} \sum_{m=1}^{M} f(\mathcal{X}_i^{(m)}; \theta)$,
92 and define the consistency term inline as $\mathcal{L}_{\text{cons}}^{(i)}(\theta) = \lambda_{\text{cons}} \sum_{m=1}^{M} \big\| f(\mathcal{X}_i^{(m)}; \theta) - y(\mathcal{X}_i) \big\|_1$. where

Table 1: Top-1 accuracy comparisons on corrupted UCF101 under continual test-time adaptation (no model reset). Results for SSv2 are provided in the supplementary material.

| Arch | Method | G | P | S | T | Z | I | D | M | J | C | R | H | Avg |
|------|--------|---|---|---|---|---|---|---|---|---|---|---|---|-----|
| | | | | | | | | | | | | | **UCF101** → | |
| TANet | No Adapt | 17.7 | 20.1 | 16.2 | 30.5 | 39.6 | 35.6 | 36.3 | 38.6 | 43.7 | 45.6 | 48.8 | 51.6 | 35.4 |
| | NORM [19] | 10.1 | 14.7 | 12.9 | 25.3 | 33.7 | 30.3 | 30.7 | 33.5 | 36.4 | 38.5 | 42.1 | 45.4 | 29.5 |
| | DUA [8] | 39.9 | 24.5 | 22.2 | 24.8 | 19.9 | 23.5 | 20.4 | 17.9 | 16.2 | 14.8 | 13.6 | 12.6 | 20.8 |
| | TENT [22] | 10.4 | 9.8 | 8.4 | 17.4 | 23.8 | 21.4 | 23.0 | 25.4 | 27.9 | 29.3 | 32.5 | 35.3 | 22.1 |
| | SHOT [16] | 10.4 | 14.9 | 13.2 | 25.7 | 33.1 | 29.9 | 30.0 | 32.3 | 34.3 | 36.1 | 39.8 | 42.9 | 28.6 |
| | T3A [6] | 16.5 | 20.4 | 17.2 | 31.2 | 39.1 | 35.3 | 34.5 | 36.5 | 41.7 | 43.7 | 46.9 | 49.6 | 34.4 |
| | ViTTA [10] | 44.3 | 50.5 | 50.2 | 59.5 | 63.9 | 63.9 | 63.6 | 64.7 | 67.3 | 68.3 | 70.2 | 71.1 | 61.5 |
| | **WSA2D** | 60.8 | 65.2 | 62.7 | 69.4 | 72.2 | 71.3 | 69.3 | 70.2 | 72.3 | 73.1 | 74.7 | 75.2 | 69.7 |
| | **WSA3D** | 60.1 | 65.5 | 64.5 | 70.5 | 73.6 | 73.2 | 71.8 | 72.7 | 74.6 | 75.1 | 76.5 | 76.9 | 71.3 |
| Video Swin | No Adapt | 71.2 | 70.3 | 63.8 | 70.5 | 72.8 | 72.8 | 73.4 | 74.2 | 75.2 | 76.0 | 77.2 | 78.0 | 72.9 |
| | TENT [22] | 75.6 | 74.1 | 68.4 | 74.2 | 75.9 | 75.8 | 76.0 | 76.6 | 77.5 | 78.4 | 79.4 | 80.0 | 76.0 |
| | SHOT [16] | 71.2 | 70.3 | 63.8 | 70.5 | 72.8 | 72.8 | 73.4 | 74.2 | 75.2 | 76.0 | 77.2 | 78.0 | 72.9 |
| | T3A [6] | 71.2 | 70.3 | 64.8 | 71.3 | 73.4 | 73.6 | 74.3 | 75.0 | 76.1 | 77.2 | 78.4 | 79.1 | 73.7 |
| | ViTTA [10] | 75.6 | 76.1 | 73.3 | 78.3 | 79.2 | 78.9 | 77.8 | 78.3 | 79.3 | 80.3 | 81.4 | 82.0 | 78.4 |
| | CMAE [11] | 63.9 | 61.6 | 54.1 | 62.3 | 65.3 | 65.3 | 65.8 | 66.8 | 68.1 | 69.1 | 70.6 | 71.6 | 65.4 |
| | ViDA [12] | 65.8 | 66.8 | 64.5 | 71.6 | 72.6 | 73.5 | 73.6 | 73.7 | 74.8 | 75.9 | 77.0 | 77.7 | 72.3 |
| | REM [4] | 65.2 | 63.1 | 57.3 | 64.7 | 67.4 | 67.9 | 68.0 | 69.0 | 70.3 | 71.3 | 72.7 | 73.6 | 67.5 |
| | **WSA2D** | 76.9 | 77.4 | 74.9 | 79.6 | 80.1 | 79.9 | 79.4 | 79.5 | 80.4 | 81.3 | 82.1 | 82.3 | 79.5 |
| | **WSA3D** | 76.6 | 75.9 | 73.3 | 78.4 | 79.3 | 79.4 | 78.9 | 79.4 | 80.1 | 81.2 | 82.3 | 82.5 | 79.0 |

$\lambda_{\mathrm{cons}} = 0.1$. Gradients are propagated through both the per-view predictions and the mean $y(\mathcal{X}_i)$ (i.e., no stop-gradient is applied to $y$), so the consistency term couples all $M$ views during backpropagation.

Thus, at each iteration, training minimizes the combined loss inline as $\min_\theta \left( \mathcal{L}_{\mathrm{WSA}}^{(i)}(\theta) + \mathcal{L}_{\mathrm{cons}}^{(i)}(\theta) \right)$.

# 3 Experiments

## 3.1 Datasets

We evaluate on UCF101 [20] and Something-Something v2 (SSv2) [3]. UCF101 has 13,320 videos (101 classes); we report split-1 results (9,537 train / 3,783 val). SSv2 has 168K train / 24K val videos over 174 classes. Following [10], we use level-5 severity across 12 families spanning sensor/digital noise, blur, weather, contrast, and codecs: G (Gaussian), P (Pepper), S (Salt), T (Shot), Z (Zoom), I (Impulse), D (Defocus), M (Motion), J (JPEG), C (Contrast), R (Rain), H (H.265, abr). For a deterministic continual stream, we partition the test set into contiguous blocks following the fixed curriculum G→P→S→T→Z→I→D→M→J→C→R→H (uniform split, remainder to earliest blocks), so corruption changes occur only at block boundaries.

## 3.2 Implementation Details

Backbones: TANet [14] (ResNet50 [5]) and Video Swin Transformer (Swin-B) [13, 15]. Adaptation aligns normalization-layer feature statistics [10]; losses are attached to the last two blocks (e.g., layers 3–4 in TANet). By default we optimize all parameters (SGD), use batch size 1, take one adaptation step per video, and evaluate online [2, 21, 24]. We report stream-wide top-1 accuracy. LRs: TANet $5 \times 10^{-5}$ (UCF) / $1 \times 10^{-5}$ (SSv2); Swin $1 \times 10^{-5}$. Temporal augmentation follows [10] with uniform frame sampling, random crops, and $M=2$ views. Baselines: NORM [19], DUA [8], TENT [22], SHOT [16], T3A [6], ViTTA [10] (official ViTTA code); CMAE [11], ViDA [12], REM [4] (our reimpl. on Swin, official code). We used single AMD Instinct MI200 GPU.

## 3.3 Results

### 3.3.1 Comparison to State-of-the-Art

Table 1 reports UCF101 results across 12 corruption families for both a convolutional (TANet) and a transformer (Video Swin) backbone. WSA yields consistent gains over baselines: WSA3D is strongest with TANet, while WSA2D is best with Video Swin, with WSA3D competitive and leading on other corruption types. Results on SSv2 are provided in the supplementary material. Random-shift evaluates single-step adaptation with no prior context. WSA still outperforms non-adaptive and TTA baselines (Table 2), indicating fast, stable updates from moment alignment with EMA. The backbone

Table 2: Top-1 accuracy (%). Each video is on a single randomly selected shift from a set of 13 conditions (the original test set and 12 corruption types). Results are averaged over 3 runs. Red means highest value per architecture while blue means 2nd highest value per architecture.

| Method | Venue | TANet | | Video Swin | |
|---|---|---|---|---|---|
| | | UCF101 | SSv2 | UCF101 | SSv2 |
| No Adapt | - | 55.41 | 26.93 | 79.62 | 44.31 |
| NORM [19] | NeurIPS'20 | 33.32 | 10.63 | - | - |
| DUA [8] | ICML'20 | 41.94 | 12.53 | - | - |
| TENT [22] | ICLR'21 | 31.32 | 10.68 | 81.35 | 44.58 |
| SHOT [16] | CVPR'21 | 32.91 | 9.02 | 78.66 | 32.93 |
| T3A [6] | NeurIPS'21 | 53.32 | 24.74 | 81.02 | 43.54 |
| ViTTA [10] | CVPR'23 | 66.94 | 32.87 | 83.11 | 46.32 |
| CMAE [11] | CVPR'24 | - | - | 74.57 | 40.95 |
| ViDA [12] | ICLR'24 | - | - | 81.19 | 36.57 |
| REM [4] | ICML'25 | - | - | 76.13 | 42.91 |
| **WSA2D** | Ours | 73.12 | 33.94 | 84.86 | 42.98 |
| **WSA3D** | Ours | 71.93 | 34.09 | 84.42 | 44.83 |

Table 3: Ablation on using precomputed training statistics from different datasets. Average top-1 accuracy (%) across all corruptions for WSA3D adaptation. TANet is evaluated with various train (clean) – test (corrupted) combinations on the corrupted UCF101 and SSv2 datasets.

| Test Set | Train Stats | Avg |
|---|---|---|
| UCF101 | No Adapt | 35.4 |
| UCF101 | SSv2 | 59.0 |
| UCF101 | UCF101 | 71.3 |
| SSv2 | No Adapt | 20.3 |
| SSv2 | UCF101 | 31.1 |
| SSv2 | SSv2 | 37.5 |

trend mirrors the continual stream: WSA2D pairs best with Swin, while WSA3D is competitive or superior on temporally structured shifts—evidence that subband-aware alignment remains stable even when the shift is unpredictable.

### 3.3.2 Training Statistics

Table 3 compares using precomputed training statistics from different datasets to no adaptation and statistics from the same dataset. Matching the training statistics to the test dataset yields the strongest results; using cross-dataset statistics still provides substantial gains over no adaptation but remains below in-domain stats. This suggests WSA benefits from source (clean) statistics while remaining transferable when only cross-dataset statistics are available.

## 4 Conclusion

We introduced Wavelet Subband Alignment (WSA), a parameter-free CTTA plugin that aligns per-channel means and variances of base features and wavelet subbands via online EMAs. Applying 2D/3D DWT to normalization-layer features (WSA2D/WSA3D) enables stable, one-step updates without architectural changes or reconstruction. On UCF101 (main) and SSv2 (supp.), across 12 corruptions and two backbones (TANet, Video Swin), WSA consistently surpasses strong TTA/CTTA baselines under continual and random shifts, and remains effective with cross-dataset source statistics. Analyses show WSA shifts energy from coarse low to informative higher frequencies, sharpening features; future work includes adaptive cross-band weighting, alternative transforms/groupings, and theory for non-stationary streams.

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

# A    Technical Appendices and Supplementary Material

## A.1    Rationale for 3D subband grouping.

The 3D grouping $\{s_1, s_2, s_3, s_4\}$ pairs each spatial orientation with both temporal passbands—$s_1 = \{LLL, HLL\}$, $s_2 = \{LLH, HLH\}$, $s_3 = \{LHL, HHL\}$, $s_4 = \{LHH, HHH\}$—so that bands share the same $(H, W)$ filtering while differing only in the temporal filter. This has three practical benefits for continual test-time adaptation: (i) stability and parameter efficiency, since aligning four bands avoids introducing eight separate $\lambda_s$ coefficients (as seen in the per-iteration loss equation (6)) that would require finer tuning and extra computation; and (ii) semantic coherence with corruption modes—spatial artifacts (e.g., blur/noise/compression) predominantly affect spatial high frequencies, while temporal disturbances (e.g., motion/compression jitter) modulate the temporal passband—so grouping by spatial orientation and marginalizing over temporal low/high preserves sensitivity to corruption-relevant directions while smoothing purely temporal variance. Importantly, when the temporal axis is marginalized, the 3D bands reduce to the canonical 2D DWT subbands (LL/LH/HL/HH), ensuring consistency with the 2D case used in WSA2D.

## A.2    Diagram of SWA.

Figure 2 sketches SWA at a glance: normalization-layer features are decomposed by a 2D/3D DWT into a base branch and four subband groups, and per-channel moments (mean/variance) are aligned to offline source statistics via an online EMA. Treating the previous EMA as fixed yields lightweight, one-step updates without backpropagation-through-time or architectural changes. WSA2D addresses spatial drift while WSA3D additionally covers temporal drift; optional temporal augmentation adds a simple consistency regularizer. Overall, the design is parameter-free, reconstruction-free, and aimed at stable streaming adaptation.

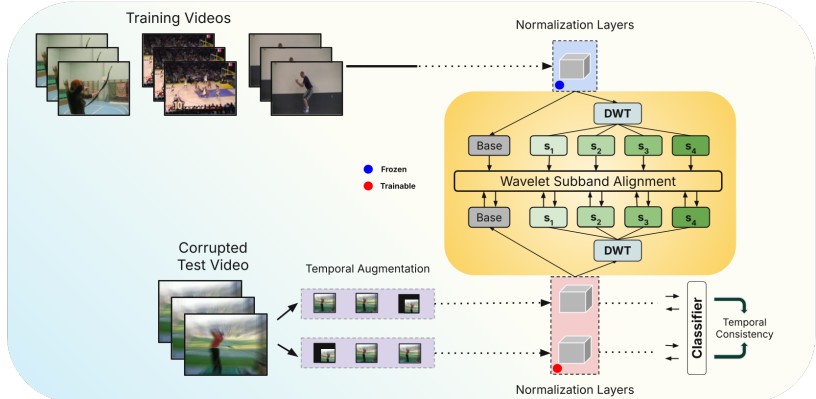

Figure 2: Overview of Wavelet Subband Alignment (WSA). Features tapped from normalization layers are transformed by a 2D/3D discrete wavelet transform (DWT) into four subband groups $s_1 - s_4$ (here $s \in \{s_1, s_2, s_3, s_4\}$). The alignment module matches per-channel first/second moments of the base and each subband to their precomputed source (clean) statistics via online exponential moving averages (EMAs). **These source/training statistics are computed only once and offline then reused for all test-time updates.** Temporal augmentation creates multiple views of each clip and a temporal consistency loss regularizes the classifier across views. Blue nodes are frozen, red nodes are trainable.

### A.3 Results on Something-Somethingv2.

Table 4 reports SSv2 results across 12 corruptions for TANet and Video Swin: WSA3D is generally strongest on temporally structured shifts, while WSA2D remains competitive, with both variants outperforming non-adaptive and TTA/CTTA baselines under continual evaluation.

Table 4: Top-1 accuracy comparisons on corrupted SSv2 under continual test-time adaptation (no model reset). *Abbreviations: G=Gaussian noise, P=Pepper noise, S=Salt noise, T=Shot noise, Z=Zoom blur, I=Impulse noise, D=Defocus blur, M=Motion blur, J=JPEG compression, C=Contrast change, R=Rain, H=H.265 bitrate reduction (abr). Red indicates the highest value per architecture; blue indicates the second highest.*

| Arch | Method | G | P | S | T | Z | I | D | M | J | C | R | H | Avg |
|---|---|---|---|---|---|---|---|---|---|---|---|---|---|---|
| | | | | | | | Something-Something V2 | | | | | | | |
| TANet | No Adapt | 13.3 | 14.6 | 12.0 | 17.4 | 23.3 | 21.8 | 23.0 | 23.3 | 23.2 | 23.2 | 25.0 | 24.1 | 20.3 |
| | NORM [19] | 8.3 | 8.6 | 7.3 | 9.9 | 12.5 | 11.8 | 12.3 | 12.4 | 13.6 | 12.9 | 13.9 | 13.7 | 11.4 |
| | DUA [8] | 7.7 | 5.8 | 4.9 | 6.4 | 5.5 | 6.1 | 5.3 | 4.8 | 4.7 | 4.3 | 4.2 | 3.9 | 5.3 |
| | SHOT [16] | 7.2 | 8.1 | 7.1 | 10.0 | 13.3 | 12.5 | 13.3 | 13.6 | 14.9 | 14.4 | 15.4 | 15.2 | 12.1 |
| | T3A [6] | 14.0 | 15.5 | 12.6 | 17.5 | 23.0 | 21.5 | 22.3 | 22.5 | 21.6 | 21.5 | 23.1 | 22.1 | 19.8 |
| | ViTTA [10] | 25.8 | 28.2 | 27.0 | 32.5 | 35.9 | 36.0 | 37.1 | 37.5 | 38.4 | 38.7 | 39.8 | 38.4 | 34.6 |
| | **WSA2D** | 33.0 | 31.7 | 46.8 | 46.9 | 31.7 | 30.7 | 36.0 | 35.1 | 38.7 | 38.1 | 38.3 | 37.8 | 37.2 |
| | **WSA3D** | 32.7 | 33.4 | 31.8 | 36.2 | 39.0 | 38.5 | 39.5 | 39.3 | 40.0 | 40.1 | 40.8 | 39.2 | 37.5 |
| Video Swin | No Adapt | 41.2 | 37.1 | 31.9 | 38.0 | 41.6 | 41.4 | 42.6 | 42.7 | 43.4 | 43.7 | 44.1 | 42.1 | 40.8 |
| | TENT [22] | 15.9 | 15.8 | 12.6 | 19.1 | 24.6 | 23.4 | 25.8 | 27.2 | 28.4 | 28.7 | 29.0 | 27.5 | 23.2 |
| | SHOT [16] | 41.2 | 37.1 | 31.9 | 38.0 | 41.6 | 41.4 | 42.6 | 42.7 | 43.4 | 43.7 | 44.1 | 42.1 | 40.8 |
| | T3A [6] | 41.3 | 36.8 | 31.9 | 37.7 | 41.4 | 41.0 | 42.2 | 42.2 | 43.1 | 43.4 | 43.6 | 41.6 | 40.5 |
| | ViTTA [10] | 46.6 | 45.1 | 43.5 | 46.8 | 48.8 | 48.8 | 49.3 | 48.8 | 49.2 | 48.9 | 49.3 | 46.8 | 47.6 |
| | CMAE [11] | 36.7 | 32.7 | 28.0 | 34.4 | 38.2 | 37.6 | 38.8 | 39.0 | 39.8 | 40.0 | 40.6 | 38.7 | 37.0 |
| | ViDA [12] | 42.8 | 42.6 | 40.7 | 44.5 | 46.9 | 46.6 | 47.8 | 47.6 | 48.2 | 48.7 | 49.2 | 47.1 | 46.1 |
| | REM [4] | 38.8 | 36.5 | 34.4 | 39.3 | 42.4 | 42.8 | 43.9 | 44.1 | 44.0 | 44.0 | 44.5 | 42.3 | 41.4 |
| | **WSA2D** | 46.8 | 44.0 | 42.0 | 44.6 | 46.1 | 45.4 | 45.2 | 44.6 | 44.9 | 44.2 | 44.4 | 42.1 | 44.5 |
| | **WSA3D** | 46.8 | 45.0 | 43.4 | 46.6 | 48.1 | 47.8 | 47.7 | 47.3 | 47.6 | 47.2 | 47.5 | 45.1 | 46.7 |

### A.4 Sliding Window Setting.

Figure 3 visualizes TANet's top-1 accuracy under a 150-video sliding-window continual adaptation protocol. The stream follows G → clean → P → clean → S → clean → T → clean → Z → clean → I → clean → D → clean → M → clean → J → clean → C → clean → R → clean → H, with clean segments interleaved between successive corruptions; at each window, evaluation alternates

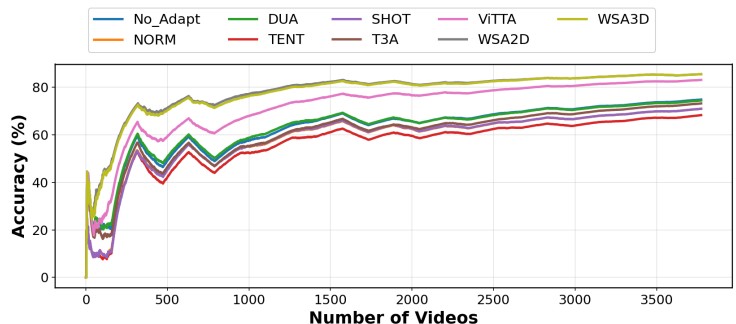

Figure 3: Top-1 accuracy (%) of TANet under continual adaptation with a sliding window. At each window, evaluation alternates between corrupted and clean UCF101 videos.

Table 5: Ablation on the effect of $\lambda_b$ and $\lambda_s$ on WSA3D using TANet with corrupted UCF101 and SSv2 datasets: Average top-1 accuracy (%). Results are shown across 12 video corruptions under continual test-time adaptation. $\lambda_s$ has four components, one for each in the band $s \in \{s_1, s_2, s_3, s_4\}$.

| Ablation | Avg Accuracy | |
|---|---|---|
| | UCF101 | SSv2 |
| No Adapt | 35.4 | 20.3 |
| $\lambda_{b=0}, \lambda_{s=[1,0,0,0]}$ | 68.5 | 36.2 |
| $\lambda_{b=0}, \lambda_{s=[0,1,0,0]}$ | 28.3 | 8.3 |
| $\lambda_{b=0}, \lambda_{s=[0,0,1,0]}$ | 28.7 | 9.1 |
| $\lambda_{b=0}, \lambda_{s=[0,0,0,1]}$ | 14.5 | 2.6 |
| $\lambda_{b=1}, \lambda_{s=[0,0,0,0]}$ | 61.5 | 34.6 |
| $\lambda_{b=0}, \lambda_{s=[1,1,1,1]}$ | 70.1 | 36.5 |
| $\lambda_{b=1}, \lambda_{s=[1,1,1,1]}$ | **71.3** | **37.5** |
| $\lambda_{b=0.2}, \lambda_{s=[0.2,0.2,0.2,0.2]}$ | 60.3 | 32.6 |
| $\lambda_{b=0.5}, \lambda_{s=[0.5,0.5,0.5,0.5]}$ | 67.8 | 36.3 |

248 between the current corrupted block and the clean. The plotted trajectory, therefore, oscillates between
249 clean and corrupted evaluations as the window advances. When the stream transitions to a new
250 corruption, accuracy typically dips and then recovers as the model adapts.

251 Compared to prior methods shown in Figure 3, **WSA3D** and **WSA2D** trace the top trajectories across
252 the stream: they rebound more quickly after each corruption switch and maintain higher steady-state
253 accuracy, whereas baselines (e.g., ViTTA, T3A, TENT) exhibit deeper dips and slower recovery.

### A.4.1 Effect of $\lambda_b$ and $\lambda_s$

255 Table 5 indicates why both terms matter. Base-only corrects global bias but ignores frequency-specific
256 drift; aligning a single subband can over-constrain that band and distort the cross-band energy balance,
257 hurting recognition. Jointly weighting $\lambda_b$ and $\{\lambda_s\}$ moves energy coherently across bands while
258 anchoring the base mean/variance, yielding the strongest and most stable results. Moderate, uniform
259 weights also work well in comparison.

### A.4.2 Effect of $\mathcal{L}_{\text{WSA}}$ and $\mathcal{L}_{\text{cons}}$

261 Table 6 shows $L_1 > L_2$ for $\mathcal{L}_{\text{WSA}}$ across models, consistent with heavier-tailed robustness to
262 corruption outliers. Temporal consistency yields minimal change for WSA2D on UCF101 ($-0.1$),
263 but helps WSA3D on UCF101 ($+0.8$) and both variants on SSv2 (about $+1.0$). SSv2 is more
264 motion-dependent, so enforcing agreement across resampled clips suppresses view-specific noise;
265 many UCF101 classes are appearance-dominant, so extra temporal regularization brings little benefit
266 for 2D alignment.

Table 6: Effect of loss type and temporal consistency. Average top-1 accuracy of TANet on corrupted UCF101 and SSv2 under continual test-time adaptation. $L_1$ outperforms $L_2$ for both WSA2D/WSA3D. Adding the temporal consistency loss $\mathcal{L}_{\mathrm{cons}}$ maintains UCF101 and improves SSv2, yielding the strongest overall results.

| Method | $\mathcal{L}_{\mathrm{WSA}}$ | $\mathcal{L}_{\mathrm{cons}}$ | UCF101 | SSv2 |
|--------|-------|-------|--------|------|
| No Adapt | – | – | 35.4 | 20.3 |
| WSA2D | $L_2$ | × | 66.9 | 32.5 |
| WSA2D | $L_1$ | × | **69.8** | 36.3 |
| WSA2D | $L_1$ | ✓ | 69.7 | **37.2** |
| WSA3D | $L_2$ | × | 67.1 | 32.2 |
| WSA3D | $L_1$ | × | 70.5 | 36.5 |
| WSA3D | $L_1$ | ✓ | **71.3** | **37.5** |

## A.5 Frequency Transforms

Table 7 compares FFT, DCT, and DWT when plugged into WSA under continual test-time adaptation on TANet/UCF101. DWT consistently delivers the strongest results for **WSA3D** across corruptions and in aggregate, while for **WSA2D** the difference between DCT and DWT is small with DCT slightly ahead on average. Compared to global Fourier bases, wavelet subbanding offers localized spatial and spatiotemporal bands that better match how noise, blur, and compression appear in streaming video, aligning with the frequency-aware CTTA premise.

Implementation notes (DCT/FFT). To mimic DWT-style subbanding, we apply separable transforms on feature tensors and partition their spectra into the same four bands. For **WSA2D**, a 2D transform is applied per frame on $(H, W)$ and the unshifted spectrum is split into four spatial quadrants, mapped to $(\mathrm{LL}, \mathrm{LH}, \mathrm{HL}, \mathrm{HH})$. For **WSA3D**, a 3D transform over $(T, H, W)$ yields eight octants, which we collapse into the four bands by pairing $\{\mathrm{LLL}, \mathrm{HLL}\} \rightarrow \mathrm{LL}$, $\{\mathrm{LLH}, \mathrm{HLH}\} \rightarrow \mathrm{LH}$, $\{\mathrm{LHL}, \mathrm{HHL}\} \rightarrow \mathrm{HL}$, and $\{\mathrm{LHH}, \mathrm{HHH}\} \rightarrow \mathrm{HH}$. We use forward-normalized transforms (energy-preserving), compute per-channel means/variances over batch and space–time exactly as with DWT, and keep the same per-iteration loss: a base term weighted by $\lambda_b$ plus a sum over subbands $s \in \mathcal{S}$ weighted by $\lambda_s$.

Table 7: Comparison on FFT, DCT, and DWT WSA2D/3D on TANet with corrupted UCF101 dataset - Top-1 Accuracy (%). Continual TTA mode without model reset from G to H.

| Corruption | WSA2D | | | WSA3D | | |
|------------|-----|-----|-----|-----|-----|-----|
| | FFT | DCT | DWT | FFT | DCT | DWT |
| Gauss (G) | 46.5 | **61.1** | 60.8 | 44.0 | 59.8 | 60.1 |
| Pepper (P) | 52.2 | 64.4 | 65.2 | 50.3 | 63.3 | **65.5** |
| Salt (S) | 51.8 | 62.7 | 62.7 | 50.1 | 62.1 | **64.5** |
| Shot (T) | 61.0 | 69.5 | 69.4 | 59.6 | 69.1 | **70.5** |
| Zoom (Z) | 65.1 | 72.4 | 72.2 | 63.9 | 72.3 | **73.6** |
| Impulse (I) | 65.0 | 71.7 | 71.3 | 64.0 | 71.6 | **73.2** |
| Defocus (D) | 64.4 | 69.9 | 69.3 | 63.5 | 69.8 | **71.8** |
| Motion (M) | 65.6 | 70.6 | 70.2 | 64.6 | 70.6 | **72.7** |
| JPEG (J) | 68.1 | 72.6 | 72.3 | 67.2 | 72.6 | **74.6** |
| Contrast (C) | 69.1 | 73.3 | 73.1 | 68.2 | 73.0 | **75.1** |
| Rain (R) | 71.1 | 74.8 | 74.7 | 70.2 | 74.6 | **76.5** |
| H.265 (H) | 71.9 | 75.2 | 75.2 | 71.1 | 74.9 | **76.9** |
| Avg | 62.7 | 69.8 | 69.7 | 61.4 | 69.5 | **71.3** |

## A.6 Wavelet Filters

Table 8 compares Haar and Daubechies-2 filters within WSA. For **WSA3D**, Haar consistently performs better across corruptions and overall; its compact, orthogonal design yields stable statistics

Table 8: Ablation on wavelet filters (Haar vs Daubechies-2) for WSA2D/3D on TANet with corrupted UCF101 — Top-1 Accuracy (%). Continual TTA mode without model reset.

| | WSA2D | | WSA3D | |
| Corruption | Haar | Db2 | Haar | Db2 |
| --- | --- | --- | --- | --- |
| Gauss (G) | 60.8 | 60.4 | **60.1** | 57.6 |
| Pepper (P) | 65.2 | 65.7 | **65.5** | 61.9 |
| Salt (S) | 62.7 | 63.5 | **64.5** | 61.2 |
| Shot (T) | 69.4 | 70.0 | **70.5** | 68.3 |
| Zoom (Z) | 72.2 | 72.8 | **73.6** | 71.3 |
| Impulse (I) | 71.3 | 71.5 | **73.2** | 70.4 |
| Defocus (D) | 69.3 | 69.3 | **71.8** | 68.5 |
| Motion (M) | 70.2 | 70.1 | **72.7** | 69.4 |
| JPEG (J) | 72.3 | 72.1 | **74.6** | 71.5 |
| Contrast (C) | 73.1 | 72.9 | **75.1** | 72.3 |
| Rain (R) | 74.7 | 74.5 | **76.5** | 73.8 |
| H.265 (H) | 75.2 | 74.8 | **76.9** | 74.3 |
| Avg | 69.7 | 69.8 | **71.3** | 68.3 |

in online updates. For **WSA2D**, both filters are closely matched with minor differences, indicating that WSA is not sensitive to the specific family when only spatial subbands are aligned.

