# OpenReview forum: "Wavelet Subband Alignment for Continual Test-Time Adaptation in Video Action Recognition"
_EurIPS.cc/2025/Workshop/UPLB — Submitted to UPLB2025_

### Official Review · Reviewer_Fex8 · 2025-11-01
**Rejection due to lack of relevance to the workshop.**

**Rating:** 3
**Confidence:** 3

**Review:**

## Brief Summary

The empirical paper targeted to action recognition in video streams and offered an online adaptation (test-time adaptation; TTA) based on spatiotemporal frequency structures, with a focus on video corruption cases. Specifically, the paper's focal point was the normalization layers, and it presented alignments between offline clean training videos and online corrupted test videos. The alignment process, referred to as wavelet subband alignment (WSA), is predicated on on grouped frequency sub-bands, achieved by using discrete wavelet transforms (DWT) through online exponential moving averages (EMA). Importantly, the alignment process used a lightweight, one-step update based on the previous EMA. This update eliminates back-propagation through time and is compatible with different network architectures. The paper tested this method on two different architectures (TANet and Video Swin) on two datasets (Something-Something v2 and UCF101) and shows improvements compared to the baseline models. A multitude of experiments were incorporated, and a number of them demonstrated marked enhancements. The contributions of this paper include a wide array of experiments and the presentation of spatiotemporal frequency alignments, which may be of interest to some computer vision communities.

## Opinion and Suggestions

Although the empirical improvements presented by the paper is interesting, I **do not think it is relevant to the topic of the workshop**. In other words, it seems this paper did not directly address the empirical aspects of fairness or bias phenomena within the context of machine learning. Unfortunately, it also did not aim to help us understand pervasive, multifaceted issues related to bias in machine learning practices. The only connections I could find include that the method may apply to cases involving distribution shifts or video stream corruption. The only explicit discussion of bias only appeared in Appendix A.1. This appendix explains how two coefficients can stabilize the model's performance by considering frequency shifts across bands, in addition to fixing global bias. I believe these connections are weak.

Therefore, **I respectfully suggest rejecting this paper**. This suggestion is not based on the paper's excellence or potential great impact on the field, but rather on its alignment with the workshop. I strongly encourage the author(s) to submit to a more closely aligned venue.

In addition, I still would like to offer a few general comments about the paper itself, primarily concerning the writing of the paper:

1. Regrettably, this paper may not pave an unobstructed path for its readers. There are many acronyms at the beginning that are not defined properly. For instance, the term "EMA" is not defined when "exponential moving averages" are mentioned in the abstract. Yet, it is used in the introduction without a definition until section 2.1, line 50. I strongly believe that more well-crafted writing can substantially increase the readability of the paper.
2. For the formulas appeared in the paper, some concise, ordered descriptions with simple examples would be helpful. For instance, in Formula 1, the paper presented a multidimensional separable wavelet transform，but input $i$ and output $o$ and the meaning of the equation itself is not immediately clear without reader's effort for inference. Similarly, the paper introduced condensed notation before and after the formula, which required extra effort from the reader to understand. I believe this could be significantly improved by reorganizing the text and providing simple examples.
3. Regarding the experiments and the corresponding section in the appendix, it would be helpful to include a section on related work that provides brief introductions or reviews of the methods and datasets in Tables 2 and 3.
	1. Could you explain why this curriculum may be more effective than others with regard to Sections 3.1, as well as Tables 1 and 4? Any insights would be greatly appreciated.
	2. I was wondering if the `(SGD)` on line 109 should instead be placed after `batch size 1.`

---

### Decision · Program_Chairs · 2025-11-03

Reject